# Potential Value of Konjac Glucomannan Microcrystalline/Graphene Oxide Dispersion Composite Film in Degradable Plastics

**Yanjun Li \*, Laijun Yao, Ruina Bian, Fangjian Zhang, Xinmeng Zhao, Donglan Yong, Jia Liu, Gennian Mao and Yong Wang**

School of Food Science and Engineering, Shaanxi University of Science and Technology, Xi'an 710021, China
\* Correspondence: liyanjun@sust.edu.cn; Tel.: +86-(29)-13484534745

**Abstract:** Konjac glucomannan (KGM) is a promising bio-based material that can effectively mitigate the global petroleum-based plastic pollution exacerbated by the responses to COVID-19. This study first acidified KGM to obtain KGM microcrystals (MKGM) with a relatively low molecular mass. Next, different volumes of graphene oxide (GO) dispersions were mixed with MKGM to prepare composite films via physical cross-linking using glycerol as a plasticizer. The UV barrier capability, mechanical strength, thermal stability, and water resistance of these films were subsequently assessed. GO enhanced the tensile strength of the polysaccharide, while limiting its toughness. Thus, the tensile strength of the MKGM film improved from 7.80 MPa to 39.92 MPa following the addition of 12 mL of GO dispersion, and the elongation at break decreased from 46.31% to 19.2%. A morphological study revealed that the addition of different volumes of GO caused the composite films to exhibit various degrees of porosity and an enhanced water barrier capability. Introducing GO also improved the UV barrier capability and thermal stability of the composite film. Meanwhile, the composite films exhibited excellent degradation properties. Therefore, composite films prepared via the acidification of KGM and the incorporation of GO are suitable for extensive utilization in degradable plastics.

**Keywords:** konjac glucomannan microcrystal; graphene oxide; UV barrier capability; hydrophobic; thermal stability





## 1. Introduction

Since the end of 2019, COVID-19 has become the most severe global pandemic in a century. In response to the COVID-19 pandemic, single-use plastic products, such as gloves and masks, have become overused. Studies have shown that the COVID-19 pandemic has directly or indirectly generated eight million tons of non-degradable plastic waste [1]. This combined with the already severe global plastic pollution problem has posed a significant threat to human health and natural environments. To address this crisis, the consumption of petroleum-based plastic products must be gradually reduced [2]. Natural biopolymers represent a desirable substitute for such products because of their degradability, renewability, non-toxicity, and low cost [3,4].

Konjac glucomannan (KGM) is a high-viscosity and low-calorie plant gum obtained by crushing and refining konjac bulbs [5]. The structure of the molecular chain of KGM is composed of D-glucose and D-mannose bound together by a β-1-4 glycosidic bond [6]. Due to its excellent water retention capacity [7], gelation ability [8], and biodegradability [9], KGM has already been employed in degradable plastics and biomedical instruments [10,11]. However, it has poor thermal stability, high viscosity, a slow swelling rate, and a poor mechanical strength, thus limiting its further applications [12]. Recently, the capabilities of KGM have been improved by combining it with other substances. Related to this, montmorillonite (MMT) was incorporated into KGM, and glycerol was used as a plasticizing additive. The mechanical properties, transmittance, and UV absorption of the resulting MMT–KGM–glycerol composite film were significantly improved compared with those of

a pure KGM film [13]. In addition, Liu et al. [14] developed high-internal-phase Pickering emulsions (HIPEs) to stabilize the KGM. The resulting HIPEs–KGM composite films exhibited excellent oxygen penetration and mechanical strength properties. Therefore, the overall capability of pristine KGM films can be effectively enhanced by the introduction of reinforcement.

The capabilities of KGM films can also be improved via modification using methods such as deacetylation and carboxymethylation treatments, which are among the most common approaches. An increase in the degree of deacetylation was found to gradually decrease the solubility of KGM and significantly enhance the mechanical strength of the resulting film [15]. In addition, KGM nanocrystals with a decreased viscosity and molecular weight were obtained via the acidification of KGM [16]. However, there are a few studies on composites based on KGM nanocrystals.

As a reinforcing material, graphene oxide (GO) is used in combination with other substances to enhance the properties of interest. The resulting introduction of oxygen-containing groups can provide active sites for surface modification and a large specific surface area for further reactions [17]. For example, GO nanosheets (GONS) have been used to form a composite with polyvinyl alcohol (PVA). The obtained GONS/PVA composite films demonstrated good water barrier capabilities that were not observed in pure PVA films [18]. Therefore, the development of composite films using KGM as the matrix and GO as the reinforcing agent represents a valuable direction of research. The flexible natural biopolymer and the rigid sheet structure can act synergistically, leading to the significant performance enhancement of composite films.

Therefore, the purpose of this study was to mix KGM microcrystals (MKGM) with different volumes of GO dispersions to prepare GO–MKGM (GMK) composite films with excellent mechanical strength, thermal stability, water resistance, and degradation properties. This was conducted to demonstrate its great advantage as a degradable plastics, thus providing a possible solution to petroleum-based plastics pollution globally.

## 2. Materials and Methods

### 2.1. Materials

The KGM (*Mw* = 200–2000 KDa, 90%) used in this study was purchased from Hubei Consistent Konjac Biotechnology Co., Ltd. (Wuhan, China). Glycerol (analytically pure) was purchased from Tianjin Tianli Chemical Reagent Co., Ltd. (Tianjin, China). Graphite powder (chemically pure, 99.85%) and other reagents were purchased Shanghai Macklin Biochemical Technology Co., Ltd. (Shanghai, China). The dialysis bag (3500 Da) employed to produce the MKGM was purchased from Beijing Solebro Co., Ltd. (Beijing, China).

### 2.2. Preparation of GO Dispersions and MKGM

An existing method was modified to prepare GO dispersions from graphite [19].

The MKGM samples were prepared using a reported method with slight modifications [16]. First, 20 g of KGM was mixed with 250 mL of 3.2 mol/L concentrated sulfuric acid in three batches separated by 40 min. The mixture was mechanically stirred at 40 °C and 400 r/min for 7 days. After the mixing process, the MKGM suspension was cooled to 25 °C, and then washed and centrifuged several times until its pH was 3. The suspension was immediately dialyzed until the pH inside and outside of the dialysis bag was the same (pH = 5). Subsequently, the post-dialysis suspension was adjusted to a neutral pH by adding 1.0% (*w/w*) ammonia dropwise. The neutral suspension was then dialyzed again. Finally, loose MKGM was acquired using lyophilization.

### 2.3. Preparation of GMK Composite Films

First, 1 g of glycerol (optimized quantity after initial experiments) and 0 mL, 0.5 mL, 4 mL, 8 mL, and 12 mL of GO dispersion (at 2 mg/mL) were added to a three-necked flask with a certain amount of deionized water. The three-necked flask was mechanically swirled at 300 r/min for 30 min while being submerged in a 40 °C water bath. Next, 2.0 g of MKGM

was added to the solution. After ensuring that all the substances were completely mixed, the solution for the GMK composite film was obtained by shaking with ultrasonic waves until the bubbles disappeared. The prepared composite solution was uniformly applied to a glass plate (13 cm × 13 cm × 0.4 cm) with a film applicator and dried overnight in a dryer at 50 °C to form the GMK composite films. For each glass plate, 60 mL of the composite film solution was used to ensure a uniform GMK film thickness. The composite films comprising 0 mL, 0.5 mL, 4 mL, 8 mL, and 12 mL of GO dispersion were labeled MKGM, GMK-0.5, GMK-4, GMK-8, and GMK-12, respectively (Table 1). All the GMK samples were equilibrated at 25 ± 1 °C under 50% ± 2% relative humidity for 2 d to reduce the experimental uncertainty (Figure 1).

**Table 1.** Constituents of MKGM (konjac glucomannan microcrystals) and GMK (GO–MKGM) composite films.

| Sample | 1% MKGM (g) | 2 mg/mL GO (mL) | Glycerol (g) |
|--------|-------------|-----------------|--------------|
| MKGM | 2 | 0 | 1 |
| GMK-0.5 | 2 | 0.5 | 1 |
| GMK-4 | 2 | 4 | 1 |
| GMK-8 | 2 | 8 | 1 |
| GMK-12 | 2 | 12 | 1 |

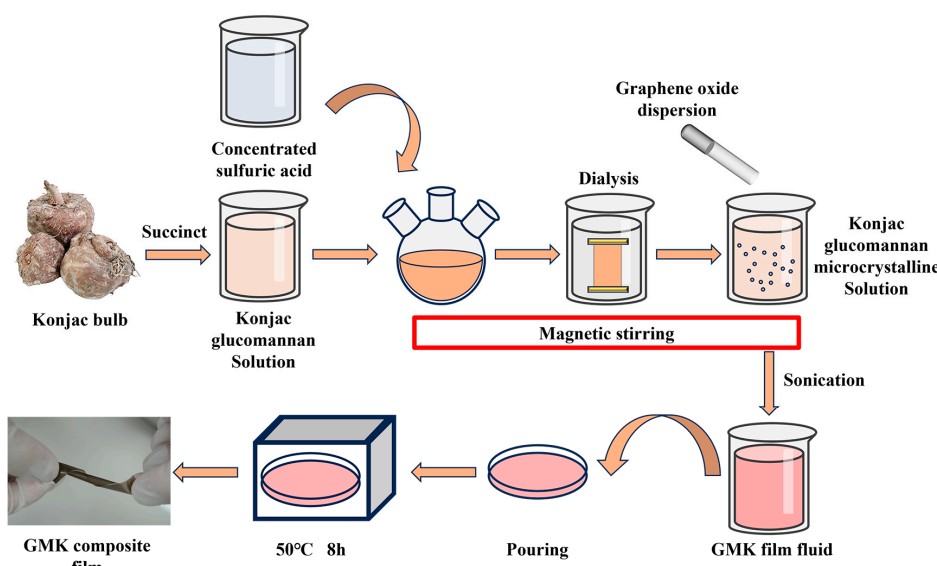

**Figure 1.** Preparation process of GMK (GO-MKGM) composite films.

### 2.4. Molecular Weight and Apparent Viscosity

KGM and MKGM solutions of the same concentration were configured, and their apparent viscosities were determined with a rotational viscosity tester (NDJ-1, Li-Chen Instrument Co., Ltd., Shanghai, China) at 25 °C. Then, their molecular weights were determined using the DNS colorimetric method [20].

### 2.5. Scanning Electron Microscope (SEM)

A 2 kV high-resolution field emission SEM (FEI-Verios-460, FEI Czech Republic Co., Ltd., Hillsboro, OR, USA) was utilized to observe the microscopic morphological features of KGM, MKGM powder, and GMK composite films. The observed brittle, fractured cross-sections were obtained by cooling the GMK samples with liquid nitrogen. All the samples were subjected to a gold spray treatment prior to observation. The surfaces of the composite films were observed at magnifications of ×2000 and ×8000, while the cross-sections were observed at a magnification of ×1500.

### 2.6. Fourier-Transform Infrared Spectroscopy (FT-IR)

The GMK composite film was ground into pieces and mixed with KBr in an agate mortar, and then compressed to form a flake. Tests were subsequently conducted using an FT-IR spectrometer (VECTOR-22, Bruker Technology Co., Ltd., Beijing, China) at 400–4000 cm$^{-1}$ with a resolution of 0.7 cm$^{-1}$.

### 2.7. X-ray Diffraction (XRD)

The samples were tested using an X-ray diffractometer (Smart-Lab-9kw, Rigaku Corporation, Beijing, China), with Cu K$\alpha$ as the radiation source. The tube voltage and current of the instrument were, respectively, set to 36 kV and 20 mA, and the scanning test was carried out over a range of 3–50° at 4°/min.

### 2.8. Thermogravimetry Analysis (TGA) and Derivative Thermogravimetry (DTG)

The heat resistances of the GMK composite films were determined using a TGA (TGA-Q500, TA., Shanghai, China). The composite film specimens used in these tests weighed 3–5 mg. The test chamber was first filled with nitrogen, and then the GMK composite film specimen was rapidly heated from 20 °C to 600 °C at 10 °C/min. The temperature precision of this test was ±0.1 °C.

### 2.9. Differential Scanning Calorimetry (DSC)

The GMK composite films were analyzed using a DSC analyzer (DSC-Q2000, TA., Shanghai, China). Each 3–5 mg composite film specimen was placed in an alumina crucible, which was subsequently sealed. The test chamber was filled with nitrogen, and then the GMK composite film specimen was rapidly heated from 20 °C to 200 °C at 10 °C/min. The protective gas flow rate in the test chamber was 20 mL/min. A separate crucible was used as a blank.

### 2.10. UV-Vis Spectra Measurement

An appropriately sized composite film specimen was clamped without folding or overlapping in a sample chamber perpendicular to the emission source for the UV-vis spectra measurement. The sample was scanned from 200 nm to 700 nm using a UV spectrophotometer (UV-2000, Rui Li Technology Co., Ltd., Beijing, China). Several MKGM films without GO were used as blanks.

### 2.11. Mechanical Properties

A servo material control tester (AI-7000-NGD, Goodtechwill Co., Ltd., Dongguan, China) was used to measure the mechanical strength of the GMK composite films. The measurement was based on GB/T 1040.3-2006 [21]. Prior to measurement, the sample film was cut to a fixed dimension with a mold, and its thickness was measured using a contact thickness gauge. The measurement rate, clamping distance, and sensor specifications of the servo material control tester were 100 mm/min, 20 mm, and 500 KGf, respectively.

### 2.12. Water Vapor Permeability (WVP)

A water vapor transmission rate tester (W3/060, Labthink Electromechanical Technology Co., Ltd., Jinan, China) was utilized to measure the water barrier capability of the GMK composite films. The measurement was based on GB/T 1037-1988 [22]. First, deionized water (10 mL) was filled into a sheet cup (Φ70 mm), which was subsequently sealed with a GMK composite film and placed in the test chamber. The test was performed at 25 °C with a relative humidity of 90%.

### 2.13. Contact Angle

An optical contact angle meter (DSA-25, KRUSS., Shanghai, China) was utilized to assess the hydrophobicity of the GMK composite film samples at an ambient temperature

of 25 °C. Ultrapure water (5 µL) was slowly dropped onto the composite film, and the contact angle was recorded after 30 s.

### 2.14. Degradation Properties of Composite Films in Soil

For the composite films, 200 mL of moist soil was collected from flower beds in Shaanxi University of Science and Technology into 500 mL beakers in a negative environment. The composite film samples were cut to the size of 1 cm × 10 cm and weighed ($W_1$). The samples were buried at the 100 mL mark of the beaker. Every 5 days, the samples were removed and rinsed, air-dried for 24 h, and weighed ($W_2$). Moreover, 20 mL of water was added to the soil to keep it moist. The final weight loss of the composite film sample was obtained.

$$\text{Weight loss} = \frac{W1 - W2}{W1} \times 100\%$$

### 2.15. Statistical Analysis

All the tests were performed three times, and the results are expressed as mean ± standard deviation (SD). One-way analysis of variance (ANOVA) and Duncan test ($p < 0.05$) were performed on the thickness, tensile strength, elongation at break, and WVP of the GMK composite films using SPSS 27.0.

## 3. Results

### 3.1. Apparent Viscosity, Molecular Weight, and Micrographs of KGM and MKGM

The microstructures of the KGM and MKGM powders are shown in Figure 2c. The KGM powder appeared in the form of irregular lumps, while the MGKM powder exhibited intertwined flakes and needle-like crystals. This morphological change occurred because the acidification and dialysis treatment caused the long-chain structure of KGM to break into short chains. To the naked eye, the MKGM solutions were more transparent than the KGM solutions were (Figure 2a). To investigate the effect of MKGM on the composite film solution, the apparent viscosity and molecular weight of the KGM and MKGM were further tested (Figure 2b). Compared with those of the KGM, the apparent viscosity and molecular weight of the MKGM are significantly lower, which is conducive to a more homogeneous dispersion of graphene oxide in the film solution, thus improving the relevant properties of the composite film.

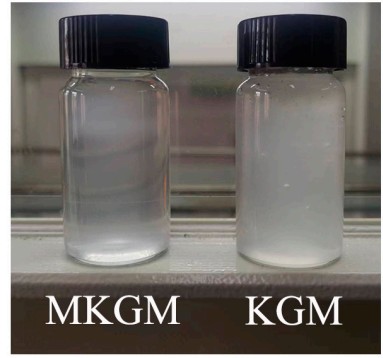

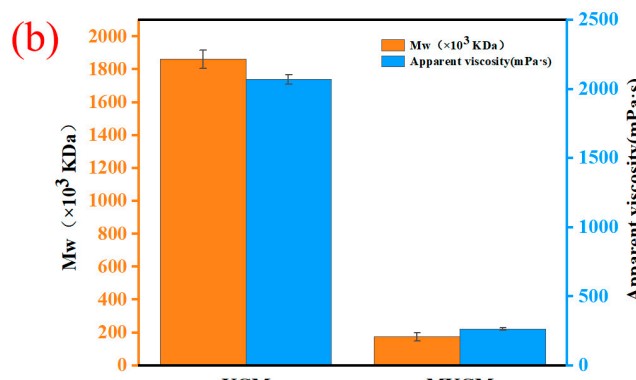

**Figure 2.** *Cont.*

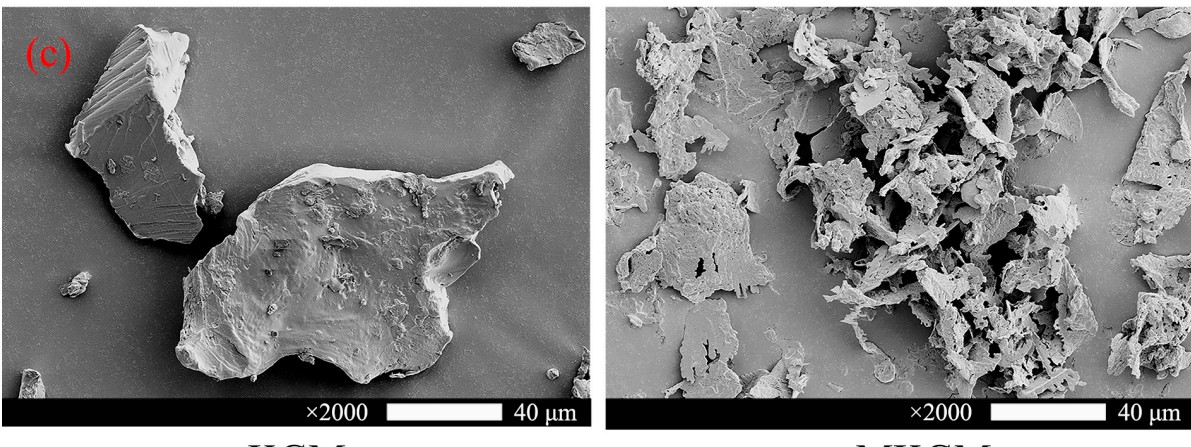

**Figure 2.** (**a**) Film solutions of MKGM (konjac glucomannan microcrystals) and KGM. (**b**) Weight-average molecular weight and apparent viscosity of KGM and MKGM. (**c**) Micrographs of KGM and MKGM powders.

### 3.2. Morphological Features of GMK Composite Films

Figure 3A shows the morphologies of the MKGM composite films by physical cross-linking them with different GO contents. The MKGM film had a uniform and smooth surface. When 0.5 mL of GO dispersion was included in the composite film, the overall morphology remained uniform, but a few folds emerged on the surface. This condition is related to the fact that the lamellar structure of GO affects the uniform distribution of the MKGM molecules. When larger quantities of GO dispersions were added, the roughness of the GMK composite films further increased, and many depressions appeared. Meanwhile, the composite films exhibited a gradual decrease in light transmittance.

SEM was used to investigate the effects of KGM acidification and different contents of GO on the microstructure of the GMK composite films. As shown in Figure 3B, the surface of the MKGM film was smooth, accompanied by minor particle accumulation caused by the incomplete dissolution of the MKGM. It is worth noting that the roughness of the composite films increased upon the introduction of GO. As GO possesses a lamellar structure, it may introduce hydrophobic groups or wrap hydrophilic groups when interacting with KGM, leading to an increase in film roughness. With the gradual increase in GO content from GMK-0.5 to GMK-4, and then to GMK-8, the composite films exhibited progressively smaller pores and more uniform porous structures, which was more conducive to the improvement of the water barrier properties of GMK composite films [23]. This may be because the presence of more GO in the composite films provided more lamellar structures to interact with the MKGM, thereby increasing the degree of physical cross-linking and further increasing the "torturous path effect" of water molecules. When infiltration occurs, GMK composite films with uniform dense lamellar structure have more tortuous migration paths, which prolongs the migration time of water and improves its water barrier capacity. However, the structure of the GMK-12 composite film was quite disorganized possibly because the excessive GO could not be uniformly dispersed in the viscous MKGM. These microscopic results agreed well with those observed by the naked eye.

A smooth and dense cross-section was observed for MKGM, whereas GMK-0.5, GMK-4, and GMK-8 all exhibited the apparent lamellar structure of GO [24]. As the GO content increased, the cross-sections exhibited greater compactness, suggesting that GO could densify the structure of the GMK composite and potentially improve the overall performance of the film. This directly explains the increased thickness of the GMK composite film with increasing GO content, which will be discussed later.

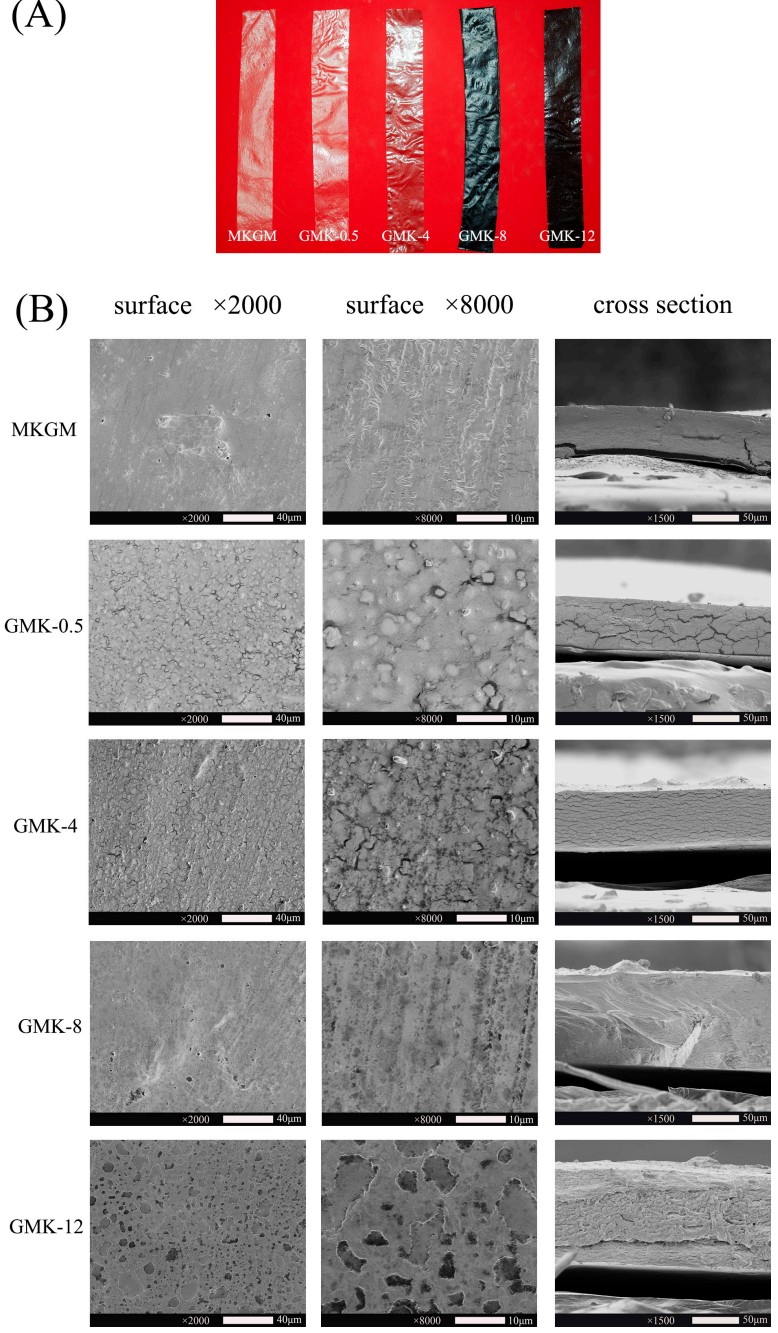

**Figure 3.** (**A**) Actual appearance photograph of MKGM (konjac glucomannan microcrystals) and GMK (GO-MKGM) composite films. (**B**) Surface and cross-section micrographs of MKGM and GMK composite films.

### 3.3. Fourier-Transform Infrared Spectroscopy of GMK Films

The physical cross-linking between MKGM and GO was verified via FT-IR (Figure 4). The absorption peaks of GO at 3429, 1720, 1400, and 1201 cm$^{-1}$ were ascribed to the O-H, C=O, C-H, and C-O-C groups, respectively, while unoxidized groups appeared at the characteristic peaks at 1629 cm$^{-1}$ (C=C vibrational absorption) and 1094 cm$^{-1}$ (C-C stretching vibration) [25]. In the spectrum for MKGM, the O-H, H-C, O-C, and C6-OH groups were associated with the characteristic peaks at 3435, 2929, 1637, and 1045 cm$^{-1}$, respectively, while the peak at 810 cm$^{-1}$ reflects the characteristic vibration of the mannose element in MKGM [26,27]. The distinct peaks of both GO and MKGM were similar to those described in previous studies. Comparing the spectra of the GO, MKGM, and

GMK composite films, the peak positions of the prominent absorption peaks in the GMK composite films remained the same as those in the pure MKGM film, and no new absorption peaks were detected. This suggests that there was no new chemical bond between GO and MKGM. The absorption peaks of O-H at GMK-0.5, GMK-4, GMK-8, and GMK-12 were at 3390, 3423, 3431, and 3398 cm$^{-1}$, respectively. Compared with the O-H peak for pure MKGM, these peaks were narrower with a red shift. A similar change was observed in the absorption peaks of the C-O group. These results suggest that a high degree of physical cross-linking occurred between GO and MKGM [28,29]. With the increase in GO, the characteristic absorption peaks of C6-OH were also red-shifted, again confirming the existence of numerous strong intermolecular hydrogen bonds after the physical cross-linking of MKGM and GO. These strong intermolecular hydrogen bonds could enhance the mechanical strength of GMK composite films, justifying the increased tensile strength, which will be discussed later in the manuscript.

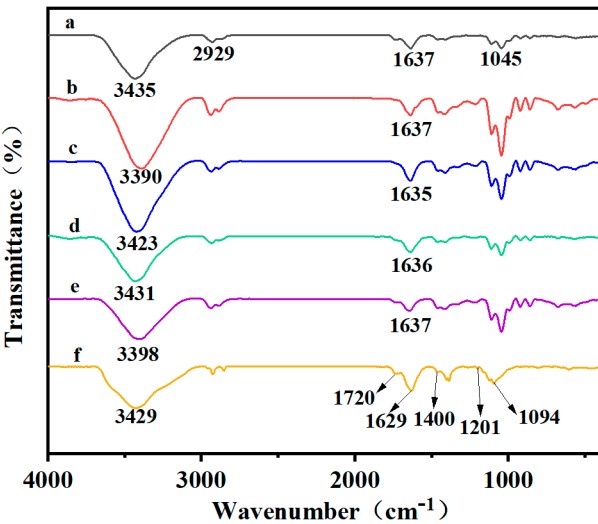

**Figure 4.** FT-IR spectra of the GMK (GO-MKGM) composite films: (**a**) MKGM (konjac glucomannan microcrystals), (**b**) GMK-0.5, (**c**) GMK-4, (**d**) GMK-8, (**e**) GMK-12, and (**f**) GO.

*3.4. XRD Analysis*

The interaction between GO and MKGM in the GMK composite films was further investigated using XRD. The pure KGM film exhibited an extremely broad amorphous peak at 2θ = 19.7° (Figure 5B), which is consistent with the previous results [30]. In contrast, the MKGM film exhibited distinct diffraction peaks at 2θ = 21.4°, 23.8°, and 26.5°, indicating a certain crystalline region. Some of the amorphous regions of KGM were destroyed during the sulfuric acid treatment, while a small portion of the crystalline regions were generated, increasing the degree of molecular ordering [31]. There were no significant changes in the peak positions for the GMK composite films with different GO contents; these positions were essentially the same as those for the MKGM film (Figure 5A). This suggests that that the interactions between different components did not produce new crystalline regions. With the increase in GO content, the diffraction peaks became sharper, and the relative intensity gradually increased, indicating that GO could improve the specific crystalline regions of GMK composite films. Therefore, as the physical cross-linking between GO and MKGM was gradually enhanced, the enhancement effect of GO in the MKGM films became stronger. This enhanced interaction was associated with the generation of numerous strong intermolecular hydrogen bonds between GO and MKGM, as suggested by the above FT-IR results.

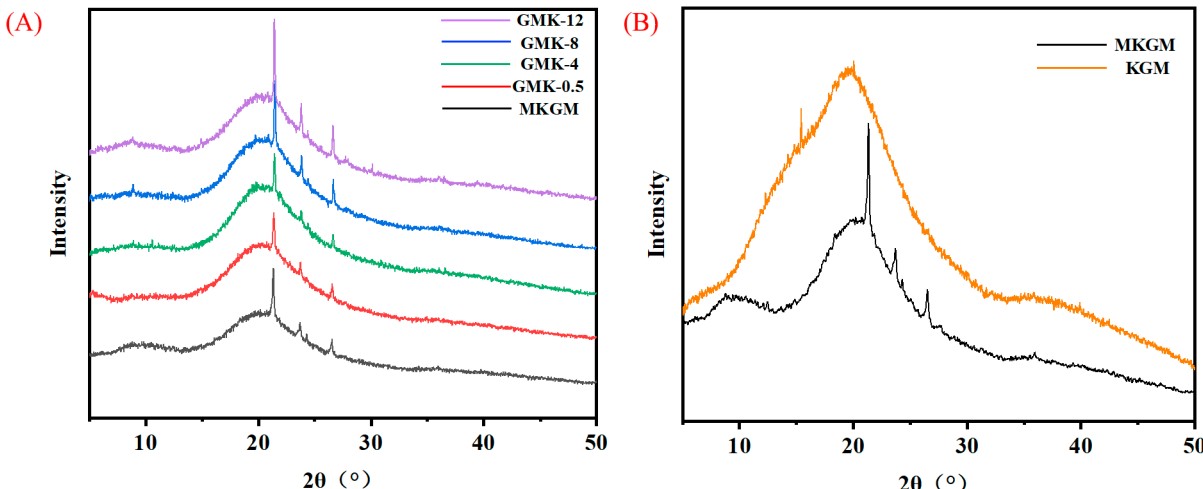

**Figure 5.** (**A**) XRD patterns of GMK-12, GMK-8, GMK-4, GMK-0.5, and MKGM (konjac glucomannan microcrystals) films. (**B**) XRD patterns of MKGM and KGM films.

*3.5. Thermal Stability Capability Analysis of GMK Composite Films*

　　The thermal stability of GMK composite films with different GO contents was evaluated via TGA at 20–600 (Figure 6A). When the temperature increased rapidly from 20 °C to 120 °C, the free water evaporated completely, and the bound water decomposed slightly, causing the GMK composite film to lose a small quantity of weight [32]. With the continued increase in temperature from 120 °C to 240 °C, the weight loss corresponds to the decomposition of glycerol in the composite film [33]. During the increase from 240 °C to 330 °C, the primary structure of the MKGM molecule disintegrated, and the sugar units were gradually broken, leading to a third stage of weight loss [34]. Eventually, when the maximum decomposition temperature was reached, the MKGM was completed carbonized, and the weight loss was significantly slow. In addition, peaks representing three temperatures were observed via DTG; the maximum weight loss rates appeared at 60 °C, 200 °C, and 310 °C, which are consistent with the TGA results (Figure 6B). The weight loss process of GMK-0.5 and GMK-4 composite films with a lower GO content was slightly delayed compared to that of the MKGM films. This may be attributed to the uniform dispersion of GO in the MKGM matrix, which led to the formation of numerous strong intermolecular hydrogen bonds between the GO and MKGM short chains, thus improving the thermal stability of GMK-0.5 and GMK-4. However, with the further increase in GO content, the weight loss process of the GMK-8 and GMK-12 composite films was instead advanced and even slightly faster than that of MKGM, which might be due to the fact that the MKGM solution still had a certain viscosity, and too much GO could not be completely dispersed. Thus, there existed a certain degree of aggregation, which was in agreement with the related studies [35]. The micrographs of the composite film just verified this statement. In addition, with the increase in GO content, the residue remaining in the composite film increased. In conclusion, the interaction between MKGM and GO improves the heat resistance of GMK composite films.

　　Additionally, DSC was further used to verify the heat resistance capability of the films (Figure 6C). The heat resistance of the GMK composite films generally increased with the GO content. The temperatures of the exothermic peaks of MKGM, GMK-0.5, GMK-4, GMK-8, and GMK-12 were 86.99 °C, 90.37 °C, 94.69 °C, 92.89 °C, and 94.89 °C, respectively. The changes in the exothermic peaks of the GMK composite films indicated that GO and MKGM developed numerous strong intermolecular hydrogen bonds during film formation [30]. Because the GMK-12 composite film exhibited the highest degree of physical cross-linking, it also exhibited the strongest heat-resistance performance, which is consistent with the TGA results.

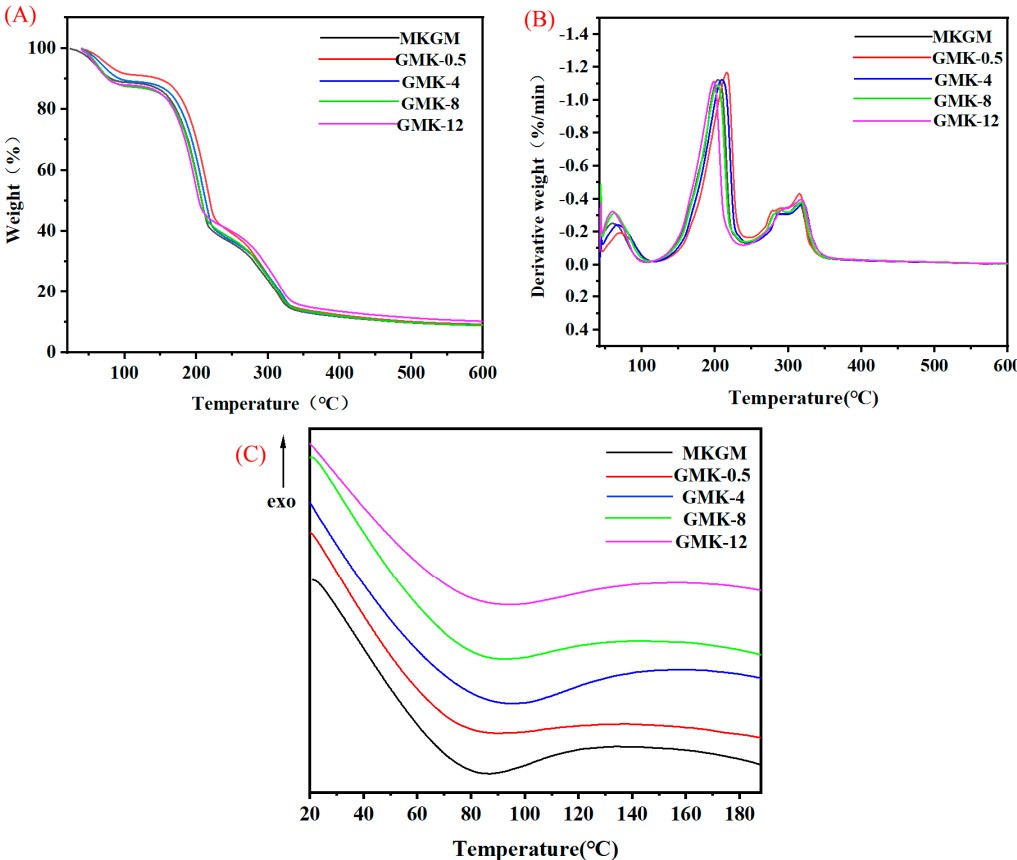

**Figure 6.** (**A**) TGA, (**B**) DTG, and (**C**) DSC curves of GMK (GO-MKGM) composite films.

### 3.6. UV-Vis Spectra Measurement of Films

The measurement of UV-vis transmission provides a critical indicator of film material performance. Figure 7 shows the transmittance of the MKGM and GMK composite films at 200–700 nm. The pure MKGM film without the presence of UV-visible light-absorbing groups exhibited high transmittance and low absorption in the UV-vis region. However, the transmittances of the composite films containing GO decreased sharply in the tested wavelength range. The UV transmittances of the GMK composite films were less than 25.6% in the 200–300 nm range and less than 2.4% in the 300–375 nm range, which has a more severe effect on the ageing of the materials [36]. This is due to the uniform distribution of GO in the film blocking the transmission of UV light. Therefore, the GMK composite films exhibited an excellent UV barrier capability, which prevents biodegradable plastics from ageing caused by direct sunlight.

### 3.7. Mechanical Strength of GMK Composite Films

Figure 8C–E shows the thickness, tensile strength, and elongation at break, respectively, of the GMK composite films. As the composite film thickness was correlated with the composition and essence of the film-forming solution [37], the introduction of GO with a rigid lamellar structure led to a slight increase in film thickness. The GO formed numerous strong intermolecular hydrogen bonds with the MKGM, changing the arrangement of the polysaccharide molecules. This improved the porosity, which further increased the water retention rate and thickness of the GMK composite films [37].

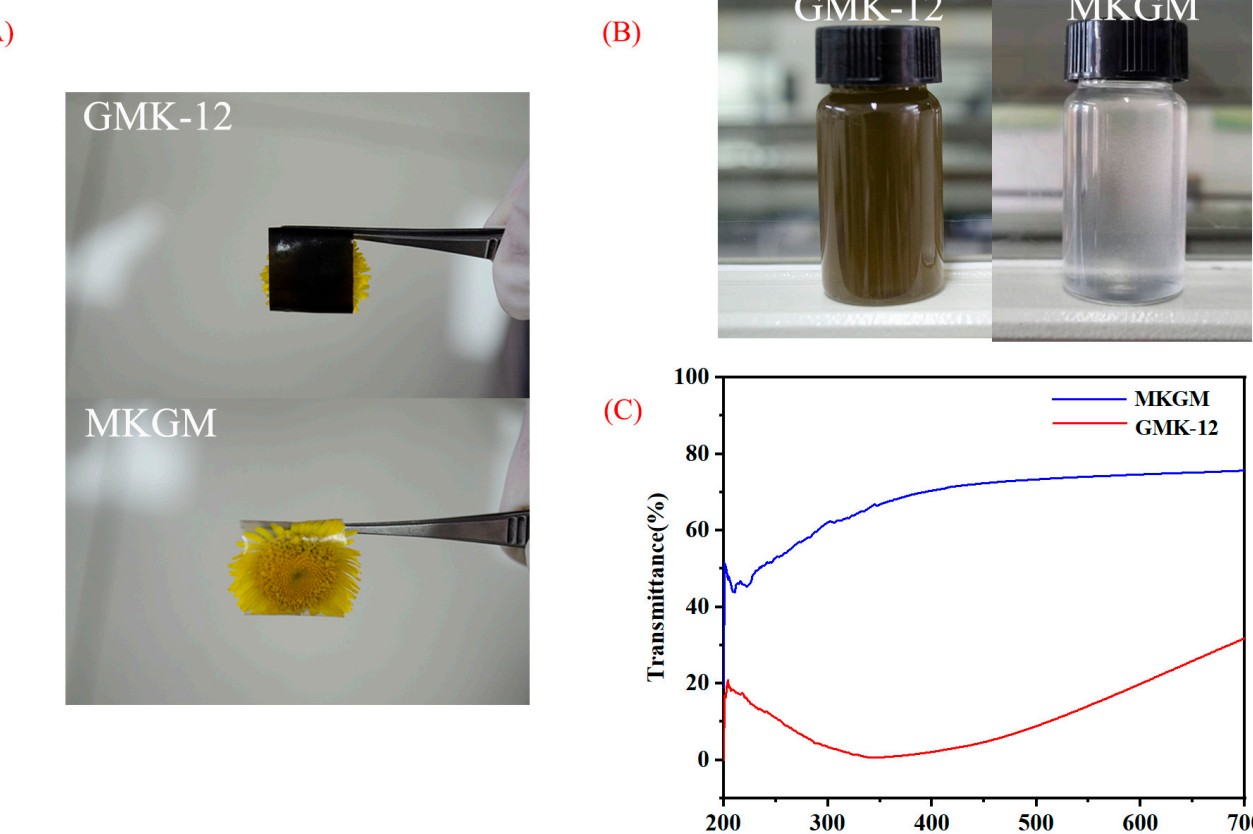

**Figure 7.** (**A**) Visual appearance of GMK-12 and MKGM (konjac glucomannan microcrystals) films. (**B**) Optical image of the composite film solution. (**C**) UV spectra of GMK (GO-MKGM) composite films.

The tensile strength was 7.801 MPa for the MKGM film, and it increased 93.4% to 15.11 MPa for the GMK-0.5 composite film. When the volume of GO dispersion reached 12 mL, the tensile strength increased to 39.92 MPa. The introduction of GO with various oxygen-containing functional groups provided active sites for surface modification and a large specific surface area for further reactions [17]. Therefore, it induced the formation of high-intensity intermolecular hydrogen bonds between the MKGM and GO to realize a significantly improved tensile strength [38]. Furthermore, adding GO to MKGM combined the advantages of the flexible macromolecules associated with natural polysaccharides with the advantages of the rigid lamellar structure of GO. This likewise enhanced the tensile strength of the composite film. In addition, the use of glycerol as a plasticizer had a similar effect on the GMK composite film.

However, the toughness of the GMK-12 composite film decreased significantly compared to that of the MKGM film, with a 58.54% decrease in the elongation at the break from 46.31% to 19.2%. This may have occurred because the rigid lamellar structure of GO limited the toughness of the polysaccharide [39]. Another possible reason for this condition is that the viscous MKGM prevented the GO from being uniformly dispersed, and the resulting GO agglomeration caused a significant attenuation of the elongation at break, which was observed with SEM observations. Therefore, the introduction of GO significantly enhanced the tensile strength, but negatively affected the toughness of the GMK composite film.

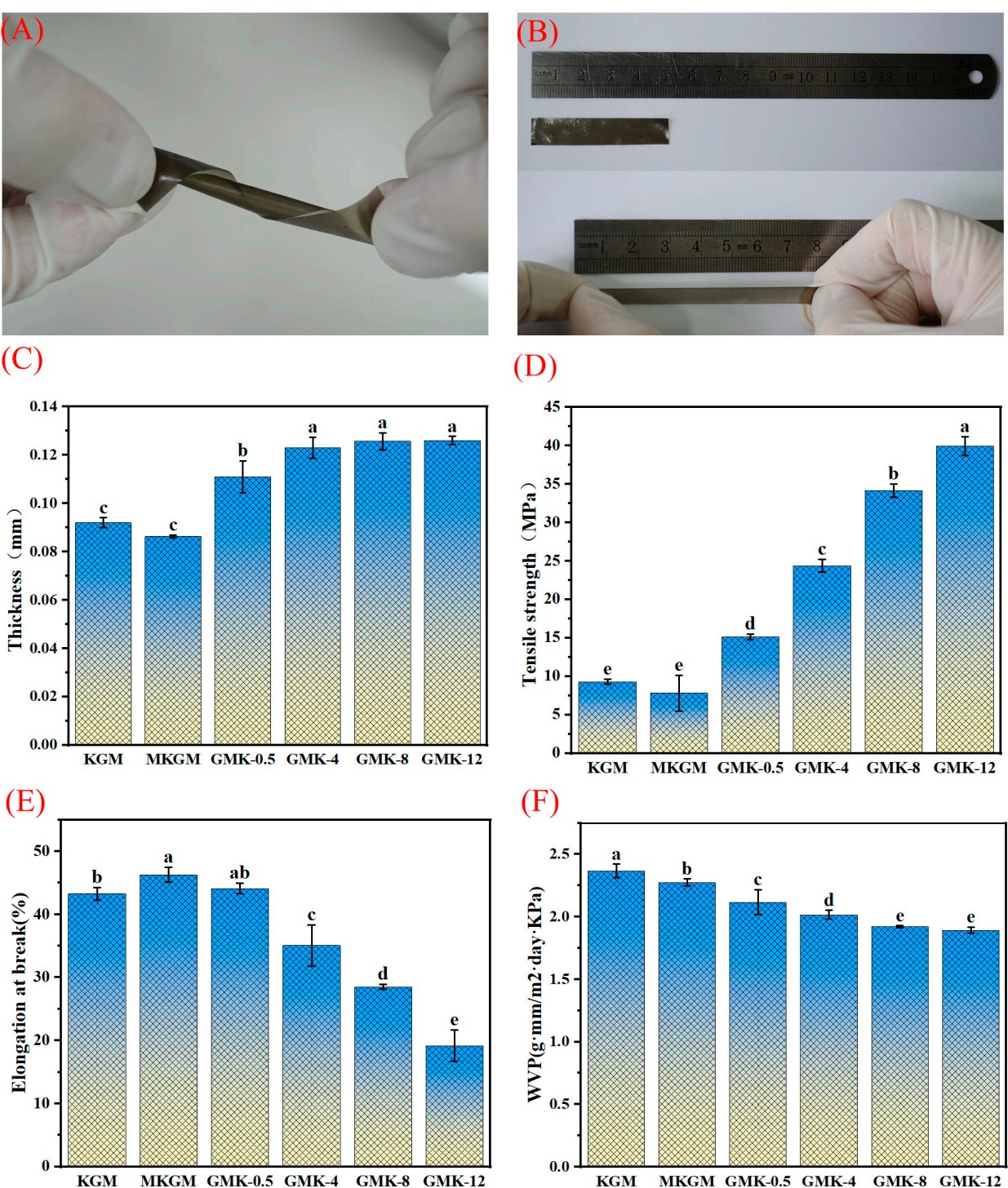

**Figure 8.** (**A**) Distortional deformability. (**B**) Tensile capability. (**C**) Thickness. (**D**) Tensile strength. (**E**) Elongation at break. (**F**) WVP of GMK (GO-MKGM) composite films. (Significance analyses were performed on the results in Figure 8 ($p < 0.05$), and lowercase letters were used to indicate significance).

### 3.8. WVP and Contact Angle of GMK Composite Films

　　The measurement of WVP provides a quantification of the water barrier capability. The WVP value is influenced by the hydrophilicity of the material, the ratio between the crystalline and non-crystalline regions, the mobility of the biopolymer chains, and the presence of pores [40]. The pure MKGM films exhibited the worst water barrier capability (WVP = 2.28 ± 0.027 g·mm/m²·day·KPa) owing to the presence of various hydrophilic



groups (Figure 8F). When GO was included, the WVP values decreased, with GMK-12 demonstrating the best water barrier capability (WVP = $1.89 \pm 0.025$ g·mm/m$^2$·day·KPa). The FT-IR and XRD results indicate that the physical cross-linking of MKGM and GO occurred, causing hydrophilic groups of MKGM molecules to be wrapped with GO, and thereby, enhancing the water barrier capability of the GMK composite film [41]. Furthermore, the SEM images showed that the pore structures of the composite films were improved by the classical lamellar structure of GO, which extended the water migration path [42], and thereby, enhanced the water barrier capability of the GMK composite films.

The wetting ability of a film's surface is typically assessed in terms of the water contact angle [43]. In general, the smaller the contact angle is, the more hydrophilic the composite film is [43]. The contact angle of the pure KGM film was the lowest (Figure 9); this corresponding high hydrophilicity was induced by hydrophilic groups, which is consistent with the WVP results obtained in this study. As the quantity of GO introduced was gradually increased, the contact angle of the GMK composite films increased owing to the interaction between the lamellar GO and low-molecular-weight MKGM, which reduced the surface polarity of the film. Therefore, the hydrophilic capability of the GMK composite film gradually decreased with the increase in GO content.

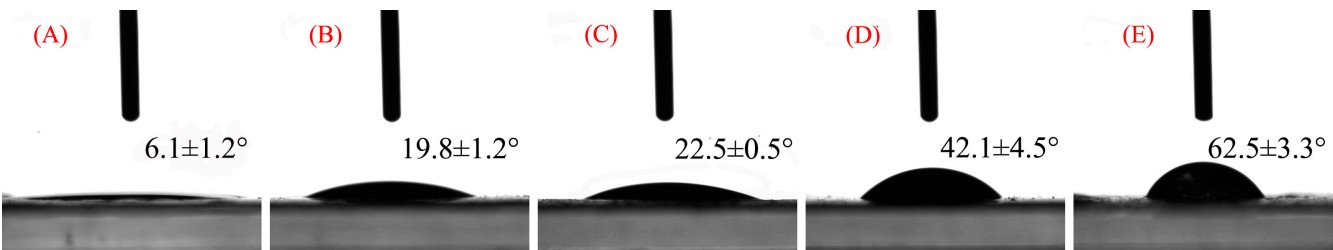

**Figure 9.** Contact angles for (**A**) the MKGM (konjac glucomannan microcrystals) films. (**B**) GMK-0.5. (**C**) GMK-4. (**D**) GMK-8. (**E**) GMK-12.

Adding a small quantity of GO to the MKGM can therefore considerably improve the water barrier capability of GMK composites and reduce the water loss, making the composite film an ideal alternative to petroleum-based plastics.

*3.9. Degradation Properties of Composite Films*

The degradation performance is one of the important indicators of the potential value of GMK composite films in plastic packaging. The samples of the composite film were buried in moist soil, and the rate of weight loss was determined at regular intervals (Figure 10). The weight loss of GMK composite film generally increased with time, and those of MKGM and KGM were as high as 60.13% and 53.47%, respectively. The faster degradation of MKGM may have occurred since MKGM with its short-chain structure is more susceptible to degradation than KGM is with its long-chain structure. The weight loss rate and speed of the composite film decreased with the increase in the GO addition. The GMK-12 composite film with the highest GO addition showed a weight loss of 52.27% after 65 days, and the degradation speed was significantly slower than that of the other composite films. The numerous strong intermolecular hydrogen bonds made GMK composite films form a dense spatial structure, which reduced the degradation performance of the composite membrane. In conclusion, the GMK composite film shows a good degradation performance in the field of plastic packaging.

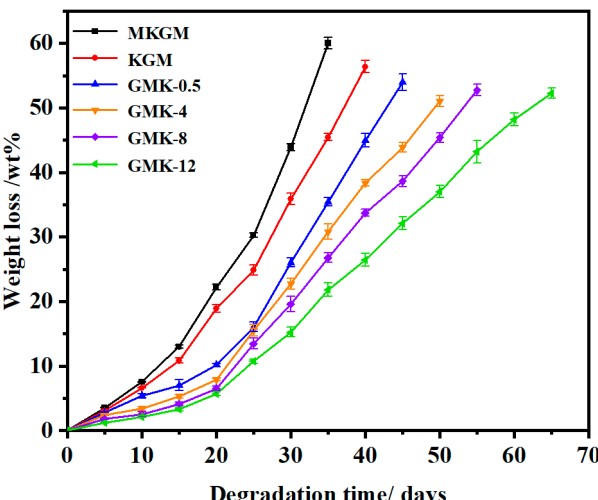

**Figure 10.** Weight loss of MKGM (konjac glucomannan microcrystals) and GMK composite films.

## 4. Conclusions

In this study, modified KGM with a reduced relative molecular mass was used as a matrix to prepare composite films. These films were evaluated to assess the effect of different GO contents on the properties of GMK composite films. The results of the FT-IR and XRD analyses revealed that the interaction between GO and MKGM generated numerous strong intermolecular hydrogen bonds that significantly enhanced the mechanical strength of the GMK composite films. Thus, compared to the MKGM films, the tensile strength of the GMK composite films exhibited an approximately five-fold increase following the addition of 12 mL of GO dispersion, while the elongation at break was reduced from 46.31% to 19.2%. The SEM observations indicated that different degrees of physical cross-linking within the GMK composite film resulted in changes in its roughness and porosity. The contact angle and WVP results demonstrated increased hydrophobicity of the composite films, which correlate with the SEM-observed change in the composite film structure. Additionally, both the UV barrier capability and thermal stability were significantly enhanced upon the introduction of GO.

In conclusion, GMK composite films with excellent mechanical properties, water barrier properties, and thermal stability are expected to replace the widely used petroleum-based plastic products, thus greatly alleviating the global plastic pollution problem caused by the responses to COVID-19. However, the antimicrobial properties of GMK composite films need to be further verified when they are applied to food packaging.

**Author Contributions:** Writing—reviewing and editing, Y.L.; Conceptualization, methodology, software, L.Y.; Data curation, writing—original draft preparation, R.B.; Visualization, investigation, F.Z. and X.Z.; Supervision, D.Y. and J.L.; Software, validation, G.M. and Y.W. All authors have read and agreed to the published version of the manuscript.

**Funding:** This work was supported by the national key research and development program (The integration and demonstration of key technologies of characteristic cash crops, fruit, and vegetable industry in Qinba Mountain area of Shaanxi Province, Project number, 2022YFD1602000); the Shaanxi Province Qinchuangyuan "scientist + engineer" team construction project (2023KXJ-239); the Shaanxi Provincial Key Research and Development Program (2023-XCZX-03, 2023GXLH-075, and 2021NY-123). The funding sources were not involved in study design, data collection, analysis, and data interpretation, report writing, or in the decision to submit the article for publication.

**Institutional Review Board Statement:** Not applicable.

**Informed Consent Statement:** Not applicable.

**Data Availability Statement:** Not applicable.

**Conflicts of Interest:** The authors declare no conflict of interest.

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
