# Peer review of "Potential Value of Konjac Glucomannan Microcrystalline/Graphene Oxide Dispersion Composite Film in Degradable Plastics"

_coatings, doi:10.3390/coatings13101757_

Round 1
Reviewer 1 Report
In this manuscript, authors reported a composite material of konjac glucomannan microcrystals and graphene oxides. Physical cross-linking between two materials and improved properties were observed from various characterization techniques. This manuscript is well structured and written in smooth language. I suggest minor revision and please find comments below.
1. The introduction emphasizes the pollution caused by disposable gloves and masks during COVID-19 pandemic. But the whole manuscript is about proposing a new promising food packaging material. This is a bit far-fetched. It might be a good idea to start the manuscript by discussing the pollution caused by food packaging.
2. The title mentioned food packaging, but the manuscript has little discussion about this. It can be helpful if author add a paragraph in introduction discussing why such materials can be used for food packaging. E.g., good strength, food contact safe, or good barrier properties?
3. In page 6-7, the sentence “the increased porosity…effectively prevented the collapse of the porous structure, extended the water migration path, and improved water barrier capability” is not clear enough. Firstly, what is the collapse of porous structure and why can it be prevented by more pores? Secondly, why water migration path is extended by more pores? Lastly, why better water barrier property can be achieved by more pores? More pores should simply open up more paths which yields poorer water barrier capability. Please revise this sentence to make it clearer.
4. In page 9, line 238, please revise the word “typologies”. Did authors mean types? topology? morphology? Or structure?
5. Page 10, line 276: The phrase “high intensity” is not accurate enough. Did authors mean the amounts of hydrogen bonds or the strength of each hydrogen bonds?
6. Page 10, line 294: The authors stated that “the weight loss process was considerably delayed in the GMK composite films.” I don’t really agree with this conclusion. From which temperature did author draw this conclusion? If at 200 °C, then it is clear that the more GO actually yields earlier weight loss. If at 310 °C, I cannot see much difference at all or maybe the red line stands out as earliest one by a little.
Minor typos such as "ageing", etc.
Author Response
Dear Professor,
Thank you greatly for taking the time to review my article and for your valuable comments.
I made the following changes in response to your suggestions.
1. I have corrected typos and grammar errors throughout the text and highlighted them in blue. In addition, the revised parts of the article have been highlighted in yellow.
2. In response to your first suggestion, I applied the composite films to biodegradable plastics, not food packaging, thereby making the logic fit the full article.
3. In response to your second suggestion, I have already discussed in the introduction,why this material can be applied to degradable plastics.
4. In response to your third suggestion, I have made the following modification: ’With the gradual increase of GO content from GMK-0.5 to GMK-4 and then to GMK-8, the composite films exhibited progressively smaller pores and more uniform porous structures, which was more conducive to the improvement of the water barrier properties of GMK composite films. This may be because the presence of more GO in the composite films provided more lamellar structures to interact with the MKGM, thereby increasing the degree of physical cross-linking and further increasing the “torturous path effect” of water molecules. When infiltration occurs, GMK composite films with uniform dense lamellar structure have more tortuous migration paths, which prolongs the migration time of water and improves its water barrier capacity’.
5. In response to your fourth suggestion, I have modified “typologies“ to ”physical cross-linking“.
6. In response to your fifth suggestion, I have modified “high intensity“ to ”numerous strong intermolecular hydrogen bonds“.
7.In response to your third suggestion, I have made the following modification: “The weight loss process of GMK-0.5 and GMK-4 composite films with lower GO content was slightly delayed compared to that of MKGM films. This may be attributed to the uniform dispersion of GO in the MKGM matrix, which led to the formation of numerous strong intermolecular hydrogen bonds between GO and MKGM short chains, thus improving the thermal stability of GMK-0.5 and GMK-4. However, with the further increase of GO content, the weight loss process of GMK-8 and GMK-12 composite films was instead advanced and even slightly faster than that of MKGM, which might be due to the fact that the MKGM solution still had a certain viscosity, and too much GO could not be completely dispersed. Thus, there existed a certain degree of aggregation, which was in agreement with the related studies. The micrographs of the composite film just verified this statement. In addition, with the increase of GO content, the residue remaining in the composite film increased. In conclusion, the interaction between MKGM and GO improves the heat resistance of GMK composite films”.
Yours,
Yanjun Li

Reviewer 2 Report
The manuscript describe the preparation of film based on GKM containing GO fro food packaging. The manuscript could be interested but the novelty should be defined before to be considered for publication.
In addition, antibacterial tests should be performed if considered for food packaging
Author Response
Dear Professor,
Thank you greatly for taking the time to review my article and for your valuable comments. I made the following changes in response to your suggestions.
1. I have corrected typos and grammar errors throughout the text and highlighted them in blue. In addition, the revised part of the article is highlighted in yellow.
2. In response to your suggestion about antimicrobial experiments. I applied the composite films to biodegradable plastics, not food packaging, thereby making the logic fit the full article. In addition, I inserted “However, the antimicrobial properties of GMK composite films need to be further verified when applied to food packaging.”in the conclusions.
3. There is some truth to what you think about the novelty of the article, but I think the topic of the article fits the journal very well.
Yours,
Yanjun Li

Reviewer 3 Report
This manuscript is written in a good manner, however, there are some points that should be considered by the authors prior to possible publication:
Comments to the authors:
- Please add your reason for selecting the GO levels (Table 1).
- Please do not use abbreviations in the Table and Figure captions.
- Please cite Figure 1, Figure 2a, Figure 5B, and Table 1 in the text.
- How many replications were used in experiments?
- Before the "Results and Discussion", a section subtitled "statistical analysis" is required in order to explain the data analysis methods or software.
- In Figure 8: Show statistically significant differences with a different letter.
- Please add some suggestions for future trends at the end of "Conclusion".
- Please only cite the important references.
Minor editing of English language required
Author Response
Dear professor,
Thank you greatly for taking the time to review my article and for your valuable comments. I made the following changes in response to your suggestions.
1. I have corrected typos and grammar errors throughout the text and highlighted them in blue. In addition, the revised parts of the article have been highlighted in yellow.
2. In response to your first suggestion, agglomeration occurs when graphene oxide dispersions are added above 12ml. So we are just doing a simple gradient division for no particular reason.
3. In response to your second and third suggestion, I have cited the figures and modified the captions of the table and figures.
4. In response to your fourth, fifth and sixth suggestion, the experiment was repeated three times and analysed for significant differences with spss 27.0.
5. In response to your seventh suggestion, I have added suggestions for future trends in the conclusion section.
6. In response to your eighth suggestion, I have rearranged the references.
Yours,
Yanjun Li

Round 2
Reviewer 3 Report
- I can see that the authors have corrected the manuscript considering all reviewers’ comments. I have no further comments on this manuscript. In my opinion, it may be allowed to be published in this Journal
Author Response
Dear professor,
Thank you greatly for taking the time to review my article and for your valuable comments.
I made the following changes in response to your suggestions.
1.The revised parts of the article have been highlighted in yellow. In addition, I replaced Figure 7.
Yours,
Yanjun Li
